Learner question’s correctness assessment and a guided correction method: enhancing the user experience in an interactive online learning system

Pal Saurabh 1
http://orcid.org/0000-0001-9438-9309 Pramanik Pijush Kanti Dutta 1 pijushjld@yahoo.co.in
Maity Aranyak 2
Choudhury Prasenjit 1 prasenjit.chowdhury@cse.nitdgp.ac.in
1 Department of Computer Science & Engineering, National Institute of Technology , Durgapur, West Bengal , India
2 School of Computing, Informatics, and Decision Systems Engineering, Arizona State University , Tempe , AZ, USA
Aljawarneh Shadi
Electronic publication date: 2021 May 25
Publication date: 2021
Volume: 7
Electronic Location ID: e532
Received 2020 Nov 23; Accepted 2021 Apr 15
Copyright: © 2021 Pal et al.
Copyright year: 2021
Copyright holder: Pal et al.
License: This is an open access article distributed under the terms of the Creative Commons Attribution License, which permits unrestricted use, distribution, reproduction and adaptation in any medium and for any purpose provided that it is properly attributed. For attribution, the original author(s), title, publication source (PeerJ Computer Science) and either DOI or URL of the article must be cited.
License URL: https://creativecommons.org/licenses/by/4.0/

Keywords: N-gram, Tri-gram, Language model, Word ordering error, Sequential pattern, Interactive system, E-learning, Java, Soft cosine similarity

Funding: The authors received no funding for this work.

==============================
In an interactive online learning system (OLS), it is crucial for the learners to form the questions correctly in order to be provided or recommended appropriate learning materials. The incorrect question formation may lead the OLS to be confused, resulting in providing or recommending inappropriate study materials, which, in turn, affects the learning quality and experience and learner satisfaction. In this paper, we propose a novel method to assess the correctness of the learner's question in terms of syntax and semantics. Assessing the learner’s query precisely will improve the performance of the recommendation. A tri-gram language model is built, and trained and tested on corpora of 2,533 and 634 questions on Java, respectively, collected from books, blogs, websites, and university exam papers. The proposed method has exhibited 92% accuracy in identifying a question as correct or incorrect. Furthermore, in case the learner's input question is not correct, we propose an additional framework to guide the learner leading to a correct question that closely matches her intended question. For recommending correct questions, soft cosine based similarity is used. The proposed framework is tested on a group of learners' real-time questions and observed to accomplish 85% accuracy.

Introduction

Online learning systems (OLSs) have brought great advantages to all kinds of formal and informal learning modes (Radović-Marković, 2010; Czerkawski, 2016; Pal et al., 2019). Over the years, OLSs have evolved from simple static information delivery systems to interactive, intelligent (Herder, Sosnovsky & Dimitrova, 2017; Huang et al., 2004), and context-aware learning systems (Wang & Wu, 2011), virtually incorporating real-life teaching and learning experience (Mukhopadhyay et al., 2020). In today's OLSs, much of the emphasis is given on designing and delivering learner-centric learning (Beckford & Mugisa, 2010) in terms of the learning style, learning approaches, and progress of a particular learner (Dey et al., 2020).

Like every learning process, one key aspect of an OLS is interaction, which makes learning more practical and dynamic (Donnelly, 2009; Pal, Pramanik & Choudhury, 2019). But, despite the advantages, due to high cost and complexity, contents developed for OLSs have limited or no interaction. The basic (or one way) interaction is incorporated in most of the OLSs through demonstration or illustration, which can be useful for very elementary learning options like remembering and comprehending. To achieve advanced learning skills like analyzing, evaluating, creating, and applying, a higher level of interactions like discussion, hands-on experiments, exchanging views with experts, etc., are required (Sun et al., 2008). The best possible way of interaction in an OLS is to devise real-time interaction between the learner and the expert/trainer (Woods & Baker, 2004; Wallace, 2003). In the absence of audio–video based interaction, the best option is to go for a question-answer based OLS (Nguyen, 2018; Srba et al., 2019) since questions are the most natural and implacable way a human enquires about information.

Interacting with a computer through natural language and to make it interpret the meaning of the communicated text has many implicit challenges associated with human-computer interaction. Existing applications like search engines, question-answering based systems (Allam & Haggag, 2012; Sneiders, 2009), chatbots (Adamopoulou & Moussiades, 2020), etc., work over user queries to deliver the required information. Fundamentally, these systems process the input query to determine its structure and semantics to understand the intention of the query. Therefore, the correctness of the semantics of the query determines the response given by these automated systems.

Significance of the correctness of the input question in an interactive learning systems

For efficient information retrieval, most of the recommendation systems focus on improving the efficiency of the recommendation engine. But, how ever efficient the recommendation engine is, if the query itself is incorrect, the search engine will not be able to retrieve the suitable information that was actually intended by the user.

Similarly, in an OLS, while interacting, if the learner inputs an incorrect question, due to the absence of the cognitive ability of the embedded search and recommendation engine, it will try to find the learning materials against the wrong input. This will lead to inappropriate learning material recommendations, which will, in effect, dissatisfy the learner, and the purpose of the OLS will not be fulfilled. Therefore, it is important that the OLS understands the learner's actual intention when she inputs a question while interacting.

Hence, in an OLS, framing the right question in terms of grammar, word use, and semantics is an absolute requirement. But often, people frame questions incorrectly, leading to ambiguous information retrieval, which misleads learners. Generally, the following are the two reasons for an incorrect question framing:Language incompetency: The lack of expertise in communicative language may cause a learner to frame a question with incorrect grammatical structure, spelling mistakes, and the inability to use appropriate words. For instance, the non-native English-speaking people having poor knowledge of English most often find it difficult to compose questions in English. For example, a question given by such a user, “HTML in how Java”, demonstrates the incorrect framing of the question. What is being asked is not understandable. It could be the programming of HTML script through Java language, or it could be the application of Java program on an HTML page. The question lacks adequate articulation, due to which the desired meaning cannot be recognized. This makes correct parsing of the question impossible.

Lack of domain knowledge: Insufficient domain knowledge also leads to frame an incorrect question. For example, the question “how a parent class inherits a child class” is syntactically correct but semantically (or technically) incorrect. Exchanging the phrases “parent class” and “child class” would make the question correct. Ignorance or the lack of domain knowledge can reason these types of semantically incorrect framing of questions. In this case, the question might be parsed successfully, but the learner will get unintended results.

In both cases, users will not get the desired answer to their questions. Therefore, it is important to validate the correctness of the question in an interactive and question-answer based automated learning system.

Research objective

From the above discussion, we can put forward the following research objectives:How to assess if the learner’s question given as input to a query-based learning system is syntactically and semantically correct or not?

If the question is not correct, then how to handle this to improve the recommendation?

Existing solution techniques, their limitations, and research motivation

In this section, we shall investigate whether the existing methods are capable of addressing the above-mentioned research objectives.

Assessing the correctness of a question

The problem of assessing the correctness of a question can be described in terms of sentence validation and meaning extraction. To address the problem of validation and semantics, the following existing techniques can be used.

NLP: The progress in natural language processing (NLP) has led to advanced techniques that allow knowing sentence structure, but comprehending its semantics still remains a challenge. NLP processing techniques for determining the intention or semantics of a sentence include sentence parsing, phrase, and relevant words (or key terms) identification. The words thus identified are being related by the relationships like dependency, modifier, subject and object, action for inducing the meaning of the sentence. In determining the relationships among the words in a sentence, often rule-based approach is adopted. Defining the rules which understand words and the relationship and interdependency among them is a non-trivial task with limited application scope. Because it is not possible to rule in the usage and relationship of all words from the English language (Sun et al., 2007b; Soni & Thakur, 2018). And hence, understanding which word combination in framing a question is correct or incorrect is very difficult. The NLP techniques assume the placement and occurrence of words in question are implicitly correct. There is no knowledge for the words that are placed incorrectly or are missing. Because of this, NLP fails to identify if the question framing is correct or not (Cambria & White, 2014; Leacock et al., 2010).

Pattern matching: In another approach, pattern matching is used to assess the correctness of the sentence. Pattern matching, in contrast to NLP techniques, is a feasible solution that allows matching a sentence pattern from available sentence patterns to find whether the sentence is matching to existing patterns or not. This approach could suitably be applied to find whether the given question is correct or incorrect. Thus, escaping the intrinsic complexity of knowledge mapping, word by word relationship and missing word problem as found in NLP techniques. In regard to pattern matching, the application of machine learning is very successful in following up with patterns. But, its inherent limitation in learning by not considering word sequence in a sentence had put constraints in verifying a question’s rightness (Soni & Thakur, 2018). For machine learning algorithm, a question "can object be a data member of a class" and "can class be data member of an object" are same. The placement of words or the sequence of words in the sentence does not matter, but only their appearance matters. So, seemingly to a machine learning algorithm, both the sentences are the same (Kowsari et al., 2019). This raises issues where machine learning fails to interpret sentences that are meaningfully incorrect due to misplacements of words.

Addressing the incorrect question

Generally, in interactive systems such as recommendation systems and intelligent search engines, if the user enters an incorrect query, the system can autocorrect the wrong input query and searches information against the autocorrected query. Here, the user’s involvement is not required. But this approach suffers from the following issues (Wang et al., 2020):It is limited to structure and syntactic corrections of the sentence.

Not able to correct the semantic errors.

The intention of the query is not judged; hence, the correctness of the query may not be exact or appropriate.

Motivation

From the above discussions, it is obvious that the existing approaches have significant limitations in addressing our research goals. Moreover, none of the work has addressed the case of learner query in an OLS, especially the issues mentioned in the previous subsections “Assessing the correctness of a question” and “Addressing the incorrect question”. Furthermore, no work is found for checking the correctness of a learner’s question submitted to an OLS and to resolve the issue if the input question is incorrect.

Proposed solution approach

Considering the research gap, we propose the following two ways to address the two above mentioned research objectives:Using a tri-gram based pattern matching to check the sentential (by construction) and semantical (meaning) structure of the question.

Instead of autocorrecting, guiding the learner to the intended correct question through one or more turns of question suggestions.

The abstract layout of the proposed approach is shown in Fig. 1.

Figure 1 Layout of the proposed work and the implementational environment.

Authors’ contribution

To attain the above-mentioned proposals, in this paper, we made the following contributions:

a) To assess the correctness of the learners’ questions:

We built two sets of corpora comprising 2,533 (for training) and 634 (for testing) questions on core Java.

We generated a tri-gram language model.

We created a classifier to identify the correct and incorrect questions based on the tri-gram language model.

The classification is evaluated on the test corpus data.

The efficacy of the classifier was compared with other n-gram models as well as with other research works.

b) To address the issue of incorrect question:

We proposed a framework for suggesting correct questions to the learner.

We designed a web-based client/server model to implement the framework.

The efficacy of the framework is assessed by a group of learners.

The proposed similarity model used in the framework is compared with other existing similarity measures.

The performance of the framework is assessed by Shannon's diversity and equitability indices.

Paper organization

“Related Work” mentions related work discussing the different error-checking methods and their limitation. “Assessing the Correctness of the Learners Input Questions” presents the correctness assessment methodology of the learners’ questions. Guiding the learner to find the correct question is presented in “Guiding the Learner to the Probable Correct Question”. The experiments and the result analysis of both the proposed methods are discussed separately in their respective sections. “Conclusions and Further Scope” concludes the paper.

Related work

Identifying the correctness of a question is related to determining the errors in the sentential text. Sentential errors are not limited to the semantics of the text but to other different types of errors like the wrong usage of words, spelling mistakes, punctuation marks, grammatical errors, etc. Soni & Thakur (2018) categorized the errors in a sentence as:Sentence structure error: The error in a sentence generates due to different organizations of POS components in a sentence.

Spelling error: The error which is generated due to the wrong spelling of words or meaningless strings in a sentence.

Syntax error: The error in sentence due to wrong/violation of grammar. The syntax error is of the following types:Subject-verb error

Article or determiner error

Noun number error

Verb tense or verb form error

Preposition error

Punctuation error: The error in a sentence, which is generated due to misplacing or missing punctuation marks.

Semantic error: The error that makes the sentence senseless or meaningless due to the wrong choice of words and their placing.

Among these five error types, detecting sentence structure error, syntax error, and semantic errors are the significant ones for finding the correctness of a question sentence used in a query-based interactive online recommendation system. Different approaches and strategies are found in the literature for detecting the different types of errors in a textual sentence. These different error detection approaches can be categorized as a rule-based approach, statistical approach, and hybrid approach (Soni & Thakur, 2018). These different error detection categories that are adopted in some notable research work that has been carried out for detecting the significant errors in a textual sentence are shown in Table 1.

Table 1 Related work categorization based on error type and resolving approach.

Error detection approach	Working method	Sentence structure error	Syntax error	Semantic error	
Rule-based approach	The rule-based approach calls for the application of linguistic rule devised by a linguistic expert for assessing the sentence to find the errors. The rule-based approach includes NLP techniques, tree parsing, etc.	(Malik, Mandal & Bandyopadhyay, 2017; Chang et al., 2014; Tezcan, Hoste & Macken, 2016; Lee et al., 2013a)	(Malik, Mandal & Bandyopadhyay, 2017; Chang et al., 2014; Tezcan, Hoste & Macken, 2016; Othman, Al-Hagery & Hashemi, 2020)	(Chang et al., 2014)	
Statistical approach	The statistical approach uses different statistical and modelling techniques to know more about the existing patterns to infer knowledge. The statistical approach includes techniques like machine learning, pattern matching and mining.	(Ganesh, Gupta & Sasikala, 2018; Schmaltz et al., 2016; Islam et al., 2018; Xiang et al., 2015; Zheng et al., 2016; Yeh, Hsu & Yeh, 2016; Ferraro et al., 2014)	(Rei & Yannakoudakis, 2017; Ge, Wei & Zhou, 2018; Zhao et al., 2019; Yannakoudakis et al., 2017; Felice & Briscoe, 2015; Wang et al., 2014; Xiang et al., 2015; Zheng et al., 2016; Yeh, Hsu & Yeh, 2016; Ferraro et al., 2014; Sonawane et al., 2020; Zan et al., 2020; Agarwal, Wani & Bours, 2000; Maghraby et al., 2020)	(Yeh, Hsu & Yeh, 2016; Shiue, Huang & Chen, 2017; Yu & Chen, 2012; Cheng, Yu & Chen, 2014; Rei & Yannakoudakis, 2016; Rei & Yannakoudakis, 2017; Cheng, Fang & Ostendorf, 2017; Ferraro et al., 2014; Islam et al., 2018; Zheng et al., 2016; Xiang et al., 2015; Zan et al., 2020; Agarwal, Wani & Bours, 2000)	
Hybrid approach	Each of the approaches has shortcoming and advantages in comparison to each for detecting an error in the text. Since the implicit working procedure for these techniques is not competent enough to identify the errors, thus the techniques are often combined as a hybrid approach to overcome the limitation of each other.	(Sun et al., 2007a)	(Kao et al., 2019; Lee et al., 2014)		

It is seen that the rule-based approach has been quite effective in detecting sentence structure error, syntax error, and punctuation error. While, the statistical approach works well to find the structure errors, spelling errors, and semantic errors (word usage and placement error). Most of the research works for detecting an error in a textual sentence are limited to word ordering error, wrong usage of words, word collocation errors, and grammatical errors in a sentence.

The sentence structure errors due to the disarrangement of words (misplaced words) and incorrect organization of the sentence's POS components have been mitigated differently. A rule-based approach was used by Malik et al. (Malik, Mandal & Bandyopadhyay, 2017) by applying POS identification and NLP production rule to check the grammatical error in the sentence. Chang et al. (2014) proposed a rule-based database approach to detect word error, word disorder error, and missing word error. Similarly, Lee et al. (2013a) manually created a list of 60 rules to detect sentence structure errors. In another approach, Tezcan, Hoste & Macken (2016) proposed a rule-based dependency parser that queries a treebank for detecting sentence structure error. In the statistical approach, n-gram based (Ganesh, Gupta & Sasikala, 2018) and machine learning based (Schmaltz et al., 2016) techniques are followed to determine the errors. Islam et al. (2018) proposed sequence to sequence learning model which uses encoder-decoder architecture for resolving missing word error and incorrect arrangement of words in the sentence. The decoder is a recurrent neural network (RNN) along with long and short-term memory (LSTM) for decoding the correct substitute for grammatical errors. Sun et al. (2007a) followed a hybrid approach to resolve the sentence structure error. They used an NLP-based POS tagging and parse tree to determine the features of an incorrect sentence and then classified for grammatical error using the classifiers like support vector machine (SVM) and Naïve Bayes (NB).

The syntax errors are due to wrong or inappropriate use of language grammar. Over the years, different approaches (e.g., rule-based, statistical, and hybrid) have been explored in research works. For syntax error detection, rule-based techniques like the NLP production rule (Malik, Mandal & Bandyopadhyay, 2017), rule-based database approach (Chang et al., 2014), and rule-based dependency parser (Tezcan, Hoste & Macken, 2016) have been applied. Othman, Al-Hagery & Hashemi (2020) proposed a model based on a set of Arabic grammatical rules and regular expressions. Among the different statistical techniques, the use of neural networks was found very effective in determining syntax error (Zhao et al., 2019). Different advanced variations of a neural network like bi-directional RNN with bidirectional LSTM (Rei & Yannakoudakis, 2017; Yannakoudakis et al., 2017), neural sequence to sequence model with encoder and decoder (Ge, Wei & Zhou, 2018), etc., are proposed for error detection in a sentence. Sonawane et al. (2020) introduced a multilayer convolution encoder-decoder model for detecting and correcting syntactical errors. Besides neural networks, another machine learning technique like SVM (Maghraby et al., 2020) is also found to be used for detecting syntax errors. The features that are considered for learning by various machine learning approaches are prefix, suffix, stem, and POS of each individual token (Wang et al., 2014). The error detection and correction are often carried out at the individual token level of each sentence (Felice & Briscoe, 2015). Besides the rule and statistical-based approach, hybrid approaches are also followed for syntax error detection, thereby taking the advantages of both approaches. Kao et al. (2019) used NLP and statistical methods to detect collocation errors. Sentences were parsed to find the dependency and POS of every word in the sentence. Subsequently, the collocation was matched through a collocation database to find errors. Similarly, Lee et al. (2014) applied rule-based and n-gram based techniques for judging the correctness of a Chinese sentence. A total of 142 expert-made rules were used to check the potential rule violation in the sentence, while the n-gram method determines the correctness of the sentence.

The semantic error detection has largely carried out by statistical approach using techniques like n-gram methods or machine learning. The use of RNN is quite popular in semantic error detection (Cheng, Fang & Ostendorf, 2017). Zheng et al. (2016) and Yeh, Hsu & Yeh (2016) used an LSTM-based RNN to detect errors like redundant words, missing words, bad word selection, and disordered words. While, Cheng, Yu & Chen (2014) proposed conditional random fields (CRF) models to detect word ordering error (WOE) in textual segments. Zan et al. (2020) proposed syntactic and semantic error detection in the Chinese language by using BERT, BiLSTM, and CRF in sequence. Similarly, Agarwal, Wani & Bours (2000) applied LSTM neural network architecture to make an error detection classifier for detecting two types of error - syntax and semantic errors such as repeated word error, subject-verb agreement, word ordering, and missing verb. For detecting a grammatical error with a long sentence, Rei & Yannakoudakis (2016) proposed a neural sequence labeling framework. The authors found bi-directional LSTM outperforms other neural network architecture like convolution and bidirectional recurrent. Shiue, Huang & Chen (2017) claimed that among the other classifier, the decision tree yields better performance for morphological error and usage error. Yu & Chen (2012) proposed an SVM model for error detection like an adverb, verb, subject, object ordering and usage error, prepositional phase error, and pronoun and adjective ordering error. In (Xiang et al., 2015), it is found that supervised ensemble classifier – Random Feature space using POS tri-gram probability offers better performance for semantic error detection in comparison to other supervised classifiers. Ferraro et al. (2014) saw the different grammatical errors like sentence structure, syntax, and semantic errors as collocation errors. A collocation match in a corpus would able to detect collocation errors. Besides machine learning models, a statistical model based on sequential word pattern mining has been quite effective in detecting grammatical errors (Ganesh, Gupta & Sasikala, 2018). Statistical modeling and machine learning, though easy to implement, are sometimes outperformed by rule-based techniques. In (Lee et al., 2013b; Sun et al., 2007a), it is found that rule-based techniques for detecting grammatical errors yield a better result for the Chinese language.

The choice of error detection technique depends much upon the rules and science of the text language under consideration. Error detection using rule-based techniques demands human expertise in framing the rules. A language with a plethora of possibilities for sentence making leads to difficulty in framing rules to capture the different types of error. Moreover, this technique can be specific to a domain or application context and cannot be generalized.

Unlike rule-based techniques, error detection using machine learning demands a huge dataset, which may not be available for all types of application scenarios. Recently, it is found that most of the syntax and semantic error detection in the text is carried by LSTM, RNN, Sequence to Sequence modeling techniques. But these techniques require corpus with incorrect and their corresponding correct sentence data with appropriate annotation or labeling. The creation of such corpus is a non-trivial task. Moreover, the models do not generalize well. This means if a sentence in the corpus is not large enough, the source sentence for error detection may appear strange to the model. Even though a lot of work has been done in error detection in the Chinese language, but there is a huge lacking of work for semantic error detection for the English language.

Various works have been done for detecting the sentence structure, syntactical and semantic errors in a sentence, but none have been found for assessing the correctness of question framing. Questions are actually textual sentences, but the way they are interpreted in comparison to the other textual sentences requires a different approach for error checking. Comprehending a question generally requires knowing “what is being asked”, “which key concepts are involved”, and “how the key concepts are related in context to the question”. Thus, identifying the error in question framing involves issues like identifying specific ordering of the semantic words (key concepts) and identifying the verbs. The verbs and other grammatical words which relate to the key concepts orchestrate the meaning of the question. Detecting these two is important in interpreting the meaning of the question and subsequently assessing the error or wrong question framing. The characteristic features which differentiate the error checking strategy of questions from other textual sentences are given in Table 2.

Table 2 Differentiating characteristic feature of question in relation to textual sentence.

Question	Other textual sentence	
Subject domains involved is important.	The subject domain is not important.	
Presence and specific ordering of keywords (semantic words) is significant.	No significance is given to particular words and their ordering and placement.	
Verb/grammatical words which relates the semantic words carries the entire meaning of the question.	The verb and other grammatical word play important role to the whole sentence instead limiting to specific words.	

Finding or detecting an error in question leads to two possibilities for correction—(a) automatic error correction and (b) recommending correct question. The automatic error correction techniques have not reached their maturity yet. It fails to correct sentences that are complex (logical or conceptual), and furthermore, it cannot align with the intent of the learner. Mostly the automatic error correction fails to correct semantic errors.

The other possibility is recommending the correct question, i.e., suggesting the probable correct questions to the learner against the incorrect input question. This facilitates the learner to navigate through the suggested question to choose the correct question which matches her intended question.

Most of the works on question recommendation are limited to Community Question Answer (CQA), which basically recommends the unanswered question to the user to be answered correctly (Szpektor, Maarek & Pelleg, 2013). The question recommendation is made based on the learner’s dynamic interest (Wang et al., 2017), previous interest (Qu et al., 2009), expertise (Wang et al., 2017; Yang, Adamson & Rosé, 2014), load (Yang, Adamson & Rosé, 2014), user model. Besides the CQA system, the question recommendation is commonly used in a frequently asked question (FAQ) based system, where questions similar or related to user questions are retrieved and recommended from the base. For finding similar questions, cosine similarity (Cai et al., 2017), syntactic similarity (Fang et al., 2017), concept similarity (Fang et al., 2017), and TFIDF, knowledge-based, Latent Dirichllet Allocation (LDA) (Li & Manandhar, 2011), recurrent and convolution model (Lei et al., 2016) are commonly used. Despite our best effort, we did not find work on the correct question recommendation for a given incorrect question.

The only work which is close to our framework is the work done by Giffels et al. (2014). It is a question answering system developed with much focus given on the completeness of the user input question. Mostly factoid-based questions like “wh” questions and true or false questions are accepted in the system. Every time a user inputs a question, it is lexically and syntactically analyzed to find the named entities—what is being asked and what is the subject of the question. The input question strength is calculated as a score based on its completeness. If the score is high, suitable answers are recommended from the base. When the score is less than a threshold, the user is given feedback on restructuring the question, and the entire process cycle is repeated until the input score is high than the threshold. The system has the following two big shortcomings:It does not check whether the input question is correct or not. It considers only the question is complete or not.

Based on the question score, the system gives the feedback. This puts forward a big issue. If the learner lacks knowledge and language skills, she will not be able to frame logical or conceptual questions completely or correctly. This leads to different answers which the learner may not agree with.

To address the issue of checking a question’s correctness, we have proposed a methodology that is more precise and practical. Further, an automatic navigation system is proposed that allows the learner to select the correct question nearly matching to her intent.

Assessing the correctness of the learners' input questions

In this section, we present the proposed work for assessing whether the learner's input questions to the query-based learning system are correct or not.

Theoretical background

The fundamental concepts that we adopted to assess the correctness of a question are the n-gram and sequential pattern mining. The basics of these concepts are briefed below.

N-gram

The n-gram is a sequence of n items adjacent to each other in a string of tokens (text). The items in the string could be letters, syllables, or words. The size of n can be 1 (uni-gram), 2 (bi-gram), 3 (tri-gram), and so on. For example, in the string “the world is a beautiful place”, the possible bigrams are “the world”, “world is”, “is a”, “a beautiful”, and “beautiful place”. Similarly, for the sentence “a document consists of many sentences”, the word-based tri-grams will be “a document consists”, “of many sentences”. The tri-grams can also be overlapping like “a document consists”, “document consists of”, “consists of many”, and “of many sentences”. The same applies to the other higher-level n-grams.

Sequential pattern mining

The sequential pattern is a set of items that occur in a specific order (Joshi, Jadon & Jain, 2012; Slimani & Lazzez, 2013). Sequential data patterns reflect the nature and situation of data generation activity over time. The existence of frequent subsequence totally or partially ordered is very useful to get insight knowledge. These patterns are common and natural, for example, genome sequence, computer network, and characters in a text string (Mooney & Roddick, 2013).

Sequential pattern mining (SPM) is the process of extracting items of a certain sequential pattern from a base or repository (Joshi, Jadon & Jain, 2012). Additionally, it helps to find the sequence of events that have occurred and the relationship between them, and the specific order of occurrences. Formally, the problem of subsequence in SPM is described as, for a sequence is an ordered list of events, denoted < α1 α2 … αn >. Given two sequences P = < x1 x2 … xn > and Q = < y1 y2 … ym >, then P is called a subsequence of Q, denoted as P ⊆ Q, if there exist integers 1≤ j1< j2<…< jn ≤m such that x1 ⊆ yj1, x2 ⊆ yj2, …, and xn ⊆ yjn (Slimani & Lazzez, 2013; Zhao & Bhowmick, 2003).

Need for using tri-gram based pattern matching

In this section, we justified the application of n-gram pattern matching and specifically the tri-gram for assessing the correctness of a learner question.

N-gram based pattern matching for question’s correctness assessment

Typically, the faults in an ill-framed user question lie in the sentence structure (missing subject or verb/phrase error), syntactic structure (grammatical error like subject-verb agreement, error related to the article, plural, verb form, preposition), and semantic errors (incorrect usage and placement of word).

Domain-specific questions are interrogative sentences that specify entities, concepts, and relations (between themselves) in a particular sequence. The sequential pattern focuses on how the concepts and entities are related and what interrogative meaning can be inferred from them (the question intention). Word collocation, like words around the entities, concepts, relations together, makes word clusters. The link between the different word clusters in sentence subsequences would enable us to get insight into the structural and semantic aspects of a question. In this direction, pattern match for finding the correct word clusters and their sequences could be a prospective approach in the assessment of a question.

The n-gram language model allows for pattern matching and probability estimation of n-words in a sentence. The high probability of n-gram pattern similarity match could lead us to assume that n-word cluster for a subsequence in a sentence is correct for their syntactic structure and semantic composition. If the entire sentence is split into an ordered sequence of n-gram subsequences, the aggregated probability estimation of correctness for each n-gram could lead us to assume the correctness of the entire question. Hypothetically, if we consider the probability estimation of the correctness is a cumulative assessment of individual n-gram sequences in the question, then which n-gram should be chosen for the optimum result? We shall try to find the answer to this in the next subsection.

Tri-gram: the preferred choice for language modeling

In n-gram, increasing the n value would result in clustering an increased number of words as a sequence and thus decreasing the total number of subsequences in a sentence. This leads to an increase in biasness toward similarity pattern matching and thereby decreases the similarity matching probability of diverse sequence patterns. Whereas decreasing n increases the number of subsequences in a sentence, thereby increasing the probability of similarity match at smaller sentences, but fails to find cohesion among word clusters and hence decreases the probability of accuracy for the larger sentences.

A tri-gram is a perfect capture for the desired features of the sentences and, at the same time, maintaining the optimum complexity factor of the program. While resoluting the sense from a group of words in sequence, it is observed that tri-gram (given one word on either side of the word) is more effective than two words on either side (5-gram). It is also found that increasing or reducing the word on either side of a given word does not significantly make it better or worse in n-gram sequencing (Islam, Milios & Keˇselj, 2012).

Question’s correctness assessment using tri-gram approach

In this section, we present the proposed approach for assessing the correctness of the learner question using tri-gram. The method includes building a tri-gram language model that is trained to assess the correctness of a question on Java, and devising a classification method to separate correctly and incorrectly framed questions. The details are described in the following subsections.

Tri-gram language model generation

The specific procedures for generating the tri-gram based language model are explained in the following. The process flow of the language model generation is shown in Fig. 2.

Figure 2 Steps for language model generation.

Data collection and corpus preparation

The language model is designed, trained, and tested on a corpus of sentences. To build the needed corpus, we collected a total number of 2,533 questions on the various topics of Java from books (available as hardcopy and softcopy), blogs, websites, and university exam papers. We adopted both manual and automatic approaches to extract and collect the questions. A group of four experts in the Java language was involved in the manual collection of questions. For automatic extraction, we used a web crawler with a question parser. The crawler, an HTML parsing program designed in Python language, reads the webpage and spawns across other inbound webpages. Using the appropriate regular expression, the expected question sentences were extracted from the parsed pages. The returned texts were then manually verified and corrected, if required, to obtain meaningful questions.

To test the efficiency of the proposed method in rightly identifying a correct and incorrect question, we needed a set of wrong questions as well. A number of incorrectly framed questions were collected from learners' interaction with the online learning portals and institutional online learning system and questions asked by the students in the class. The incorrect questions contain grammatical errors (sentence structure and syntactic errors) and semantic errors.

The details of the question datasets are as following:Number of questions in training dataset: 2,533 (all correct)

Number of questions in testing dataset: 634

Number of correct questions in testing dataset: 334

Number of incorrect questions in testing dataset: 300

Data preprocessing for language model generation

As the collected questions consisted of many redundancies and anomalies, we preprocessed them to develop a suitable language model for questions. Text preprocessing typically includes steps like stopword removal, lemmatization, etc. Stopwords are frequently used words like “I”, “the”, “are”, “is”, “and”, etc., which provide no useful information. Removing these from a question optimizes the text for further analysis. However, sometimes certain domain-specific keywords coincide with the stopwords, removal of which may result in a loss of information from the questions. Therefore, we modified the list of stopwords by removing the domain-specific keywords from the Natural Language Toolkit (NLTK https://www.nltk.org/) stopword list to avert eliminating the required stopwords. The modified NLTK stopword list is used to remove stopwords from the questions, excluding those which are meant for the Java language.

Each question is broken down in the form of tokens using the regular expression tokenizer, which is present in the NLTK library. Each of these tokens is converted into their stem (root word) form using the Wordnet Lemmatizer to reduce any inflectional form of words. The steps for preprocessing an input question are shown in Fig. 3.

Figure 3 Typical steps for preprocessing a question.

Language modeling

The preprocessed questions are broken down into sets of distinct uni-, bi-, and tri-gram sequences. The uni-gram set is built on individual tokens in the questions. Whereas the bi- and tri-grams are formed using overlapping two- and three-token sequences, respectively, as shown in Fig. 4.

Figure 4 Generating uni-gram, bi-gram and tri-gram sequences from a question.

The respective count of each n-gram occurrence is obtained from the question corpus. Along with the count, based on the relative occurrences in the corpus, the unconditional log probabilities of each uni-gram, as represented by Eq. (1), and conditional log probabilities of each bi- and tri-gram, as represented by Eqs. (2) and (3), respectively, are calculated.

(1) P(w1)=log⁡(C(w1)C(wn))

Where wn represents the words in the corpus and c(wn) returns the count of the total number of words in the corpus.

(2) P(w2|w1)=log⁡(C(w1,w2)C(w1))

(3) P(w3|w1,w2)=log⁡(C(w1,w2,w3)C(w1,w2))

The log probabilities in Eqs. (1) and (2) allow transforming higher fractional probability values to lower ones, which are easy to be used in the computation. A sample representation of the language model is shown in Table 3. The entire language model derived from the question corpus is saved in ARPA (http://www.speech.sri.com/projects/srilm/manpages/ngram-format.5.html) format.

Table 3 Uni-gram, bi-gram and tri-gram probabilities for a question.

Unigram	Unigram probability	Bi-gram	Bigram probability	Tri-gram	Tri-gram probability	
what	0.069	what different	0.034	what different type	0.294	
different	0.007	different type	0.157	different type operator	0.117	
type	0.008	type operator	0.023	type operator use	0.333	
operator	0.006	operator use	0.067	operator use Java	0.166	
use	0.008	use Java	0.024			
Java	0.042					

Classifying correct and incorrect questions

The correctness of a question is estimated based on its syntactical and semantic aspects and accordingly is classified as correct or incorrect. The complete process of identifying correct and incorrect questions is pictorially shown in Fig. 5.

Figure 5 The flow diagram for identifying correct and incorrect questions.

Preprocessing the learners’ input questions

The input questions from the learner are preprocessed to remove the stopwords and the irrelevant words. Also, lemmatization is carried over the input question.

Probability estimation for question correctness based on the syntactical aspect

After preprocessing, the question is broken down into overlapping tri-gram sequences. Each tri-gram sequence is estimated for probability by maximum likelihood estimation (MLE) from the language model. If a tri-gram sequence of the question is not present in the language model, it will lead to zero estimation. However, though the entire tri-gram sequence may not occur in the language model, a partial word sequence, a lower-order n-gram (bi-gram) of it, could be valid. The Backoff approach (Jurafsky & Martin, 2013; Brants et al., 2007) is considered for tri-grams to take into account of sequence which counts to zero. The tri-gram sequences which estimate to zero are further estimated for their bigrams. The probability of a tri-gram is depicted in Eq. (4)

(4) P(w3|w1,w2)={c(w1,w2,w3)c(w1,w2),ifc(w1,w2,w3)>00.5×(C(w1,w2)C(w1)+C(w2,w3)C(w2)),ifc(w1,w2,w3)=0

The probability of each tri-gram ranges from 0 <= P <= 1. A higher probability refers to more correctness and higher occurrence. The entire probability of syntactic correctness of the sentence can be obtained as the addition of probability of each tri-gram in the question in Eq. (5), where k is the number of tri-grams in the question and Pi is the probability of the ith tri-gram sequence in the sentence.

(5) Esy=1k∑i=1k⁡Pi

Probability estimation for question correctness based on semantic aspect

The correctness of question semantic is assessed by estimating the validity of individual overlapping tri-gram sequences of the sentence. The validity of the tri-gram is assessed by the probability estimation of each tri-gram sequence in question found matches in the language model, as shown in Eq. (6). The semantic correctness of a question is estimated on the full similarity match of each tri-gram sequence. More the number of subsequences of the question sentence matches the language model, more is the chance of the question being semantically correct. The overlapping tri-gram sequences reflect the cohesion among words in the sentence subsequences. Thus, increasing the number of matching of the tri-gram sequences establishes a higher probability of semantic accuracy of the question. The semantic correctness of the question is calculated as the summative average of probabilities of each tri-gram sequence in the sentence is shown in Eq. (7).

(6) P(w3|w1,w2)={1,ifP(w3|w1,w2)>00,ifP(w3|w1,w2)=0

(7) Esm=1k∑i=1k⁡Pi

Classification

The correctness of a question is calculated by Eq. (8), where Esy and Esm are the probability estimates of syntactical and semantic correctness of the sentence, respectively. A syntactically correct question has Esy = 1, and Esm = 1 for semantically correct. Hence, the standard score for a correct question is 1 + 1 = 2. Thus the degree of correctness (Cd) of the question with respect to the complete correctness (i.e., 2) is assessed by adding the calculated probability estimates Esy and Esm and subtracting from 2. We considered the question is correctly structured, if Cd ≤ 20; otherwise, the framing of the question is not correct.

(8) Cd=(2−(Esy+Esm))×50

Experiment and performance evaluation for question’s correctness assessment

The evaluation of the performance measure of the proposed approach for assessing the correctness of the learner question is done on a corpus of 634 annotated questions, where 52% of questions are correctly framed. The performance of the tri-gram approach for classifying questions as correct or incorrect is measured based on the metrics: true positive, true negative, false negative, and false positive, and the performance measures: Accuracy, Precision, Recall, F1-Score, as shown in Table 4.

Table 4 Performance measures of the proposed approach.

Performance metric	Value	Performance measure	Value	
True positive	282	Accuracy	0.9211	
False positive	18	Precision	0.9400	
True negative	302	Recall	0.8980	
False negative	32	F1-Score	0.9188	

In the experiment, we attempted to distinguish between correct and incorrect questions based on the probabilistic calculation proposed by our approach. The experimental results show that our method fails to classify 50 of these questions correctly. Out of these 50 questions, 32 were correct questions but are identified as incorrect. Further analysis of these false-negative questions reveals that after preprocessing and stopword removal, the length of most of the questions is reduced to less than three. These questions fail to generate any tri-grams to perform the probabilistic calculation. So, these questions by convention get marked as incorrect. Some of these false-negative questions even belong to domains that are not present in the training dataset. As a result, the proposed method fails to identify these questions correctly. The other set of incorrectly classified questions comprises incorrect questions which are marked as correct. The false-positive questions primarily have misplaced punctuation marks which results in the structure of the incorrect question identical to the correct questions in the training set. They form tri-grams or bi-grams, which perfectly match the tri-grams or bi-grams from the language model and render a high probabilistic score for the question. A margin of 8% error shows the efficiency of the proposed approach.

The efficacy of the tri-gram model approach was compared with other n-grams. The models were trained over the same question dataset to keep the experiment bias-free. Figure 6 shows a comparison of the accuracy measures obtained for each n-gram approach over the same statistical calculation. It is evidently seen that the accuracy of tri-gram is far better than other n-grams. The accuracy decreases with the increasing value of n in n-gram. It leads to biased higher-order word sequence pattern search and fewer options for pattern assessment at lower orders. This causes restricted pattern search and a decrease in accuracy. Similarly, decreasing n leads to word sequence pattern search at lower order, which restricts the probability of correctness of the word sequences at higher orders. This typically reduces the accuracy. The comparative experiment thus concludes that the use of the tri-gram model for question assessment leads to better assessment results.

Figure 6 Accuracy comparison of the four n-gram approaches.

The result of the proposed approach is compared with the results of another similar work by Ganesh, Gupta & Sasikala (2018), in which the authors applied a tri-gram based approach to detect an error in English language sentences. Table 5 shows the result comparison in terms of four assessment metrics. From the table, it is evident that the accuracy of our proposed approach is much better. However, the precision of both approaches is the same. This establishes the true positive and true negative identification cases are better in our approach for detecting the errors and thus the correctness or incorrectness of the question sentences.

Table 5 Comparative results of the proposed approach and the solution given in (Ganesh, Gupta & Sasikala, 2018).

	Proposed approach (%)	Result of (Ganesh, Gupta & Sasikala, 2018) (%)	
Accuracy	92.11	83.33	
Precision	94.00	94.11	
Recall	89.80	80.00	
F1-Score	91.88	86.48	

Guiding the learner to the probable correct question

In the previous section (“Assessing the Correctness of the Learners’ Input Questions”), we checked if the question given as input by the learner to the query-based learning system is syntactically and semantically correct or not. If the question is not correct, we guide the learner to the probable correct question that she actually intended to ask through one or multiple steps of question suggestions. The detailed methodology and framework of the proposed work are discussed in the following subsections.

Similarity-based recommendation for mitigating incorrect learner question

Computationally auto-correcting the incorrectly framed question is one of the acclaimed ways followed in literature. But the success is limited and restricted to correcting only a few types of errors or mistakes. The typical mistakes a learner commits while articulating a question are shown in Fig. 7. For instance, inappropriate word selection may not reflect the exact intention of the learner. Similarly, insufficient keywords may not express the intended concept.

Figure 7 Typical mistakes made by the learner in a question.

In regard to these, except for grammatical and sequential ordering errors, auto-correction for other types of errors is not possible. The other way around, the problem is suggesting correct questions to the learner which are near to what she intended to ask. Suggesting correct questions which are similar to information and morphological structure to the given question could lead to having a chance that learner may found the right question which she intends to ask. Considering the information like the concepts and functional words which are used in compiling the question is best of her knowledge in the current information seeking situation, the learner could be recommended appropriate questions which are aligned to/with the information they are seeking for. Thus, suggesting correct questions in contrast to the incorrect question imposed by the learner is through similarity-based recommendation is an effective way to overcome the incorrect question problem.

Issues in similarity-based recommendation of questions

Cosine and Jaccard similarity techniques are the two text-based similarity approach which has been widely incorporated for finding similar text (Sohangir & Wang, 2017; Amer & Abdalla, 2020). But these approaches, when applied to question-based corpus for identifying similar question text, lead to the recommendation issues, as discussed in the following subsections.

Information overload

Text similarity based on word match searches for similarity for every occurring word in the source sentence-incorrect question text for an exact match in the questions present in the question corpus. The needful comparison based on matching word occurrence among the sentences returns similar text. Since the question framing is incorrect, taking a part of the entire sentence which seemingly founds to be correct and conveys the learner’s intent, could lead to a better similarity match. However, the prevailing constraint and limitations of NLP fail to analyze and identify the parts of the source sentence, which are correct as per learner intention. Failing to determine this leads to ambiguity in identifying the parts of a sentence that are to be taken correctly for similarity match. Without this knowledge, the similarity search is done for each occurring word (assuming they are correct as per the learner intent) in the question against the questions in the corpus lead to a huge set of information. For example, a learner questions on Java with incorrect word ordering and missing words like “What different are interface implement”, when runs for similarity match like Jaccard similarity on a question corpus returns a lot of information, as shown in Table 6. With this amount of information, the learner may get confused and lost.

Table 6 Similar questions returned by Jaccard similarity for the learner question “what different are interface implement”.

	Returned similar question		Returned similar question	
1	What is the need for an interface?	31	What do you mean by interface?	
2	What are the properties of an interface?	32	How interface is different from abstract class?	
3	What is interface?	33	What are the different types of applet?	
4	What are the methods under action interface?	34	Which methods of serializable interface should I implement?	
5	What are the methods under window listener interface?	35	What is an externalizable interface?	
6	What is Java interface?	36	What is vector? how is it different from an array?	
7	What are the advantages of interfaces	37	What are the methods under action interface?	
8	What interfaces is needed	38	What are the methods under window listener interface?	
9	What is an interface?	39	What is Java interface?	
10	What are interfaces?	40	What are the advantages of interfaces?	
11	What are constructors? how are they different from methods?	41	What are constructors? how are they different from methods?	
12	How interface is different from class	42	Is it necessary to implement all methods in an interface?	
13	What is an interface? how is it implemented?	43	If you do not implement all the methods of an interface what specifier should you use for the class?	
14	What are different modifiers?	44	What is difference between interface and class?	
15	Is it necessary to implement all methods in an interface?	45	What is difference between package and interface?	
16	How interface is different from abstract class?	46	What do you mean by interface?	
17	What are different comments?	47	What do you mean by interface?	
18	What are different modifiers?	48	What is the nature of methods in interface?	
19	Is it necessary to implement all methods in an interface?	49	What do you know about the file name filter interface?	
20	How interface is different from abstract class?	50	What is a nested interface?	
21	If you do not implement all the methods of an interface while implementing what specifier should you use for the class?	51	Which classes implements set interface?	
22	What must a class do to implement an interface?	52	What is the interface of legacy?	
23	What interface must an object implement before it can be written to a stream as an object?	53	What is different between iterator and listiterator?	
24	What is applet stub interface?	54	What are different collection views provided by map interface?	
25	How interface is different from a class.	55	What is comparable and comparator interface?	
26	What is an interface?	56	What will happen if one of the members in the class doesn't implement serializable interface?	
29	What is interface?	57	What is serializable interface in Java?	
30	How interface is different from class?	58	What is externalizable interface?	

Diverse information

A learner, when composing a question, intends to seek information limited to a particular topic(s). Text similarity based on word match searches for similarity for every occurring word in the source sentence for an exact match into the question corpus. For similarity measurement, weightage is given to word occurrence frequency rather than on their subject domain relevancy. No consideration is given to individual tokens belonging to a topic of a domain. Since a question is made up of functional words (noun or verb) along with concepts (domain keywords), the word match found for every functional word in the corpus leads to different questions having different topics which the learner does not intends to seek. This results in questions that are beyond the search topic boundary, leading to diversification of information. For example, the similarity search for an incomplete question like “access modifier in Java” using Jaccard similarity returns questions of different topics, as shown in Table 7. Figure 8 shows the share of the number of questions belonging to different topics for the given similarity recommendation. A large number of questions are on a different topic than that of the input question. This may put the learner in jeopardy and confusion. Conclusively, the similarity match on functional words of the source question in the corpus may result in diversification instead of convergence.

Table 7 Recommended list of question and their topic retrieved using Jaccard similarity for the incorrect input question “access modifier in Java”.

	Recommended Question	Topic		Recommended Question	Topic	
1	What are the features of java language?	Basics	52	Briefly discuss the features of java.	Basics	
2	What is the need for java language?	Basics	53	What is jvm? Explain how java works on a typical computer?	Basics	
3	How java supports platform independency?	Basics	54	List out at least 10 difference between java & c++	Basics	
4	Why java is important to internet?	Basics	55	Explain, why java is the language of choice among network programmers	Basics	
5	What are the types of programs java can handle?	Basics	56	Write a java program to accept two strings and check whether string1 is a sub string of string2 or not.	String	
6	What are the advantages of java language?	Basics	57	Explain the relevance of static variable and static methods in java programming with an example.	Class & Object	
7	Give the contents of java environment (jdk).	Basics	58	Describe the syntax of single inheritance in java.	Inheritance	
8	Give any 4 differences between c and java.	Basics	59	Name at least 10 java api class you have used while programming.	Package	
9	Give any 4 differences between c++ and java.	Basics	60	Significance of interface in java?	Interface	
10	What are the different types of comment symbols in java?	Basics	61	Do class declaration include both abstract and final modifiers?	Inheritance	
11	What are the data types supported in java?	Data type & variable	62	Do java supports operator overloading?	Operator	
12	How is a constant defined in java?	Data type & variable	63	Does java support multithreaded programming?	Thread	
13	What are the different types of operators used in java?	Operator	64	Do java has a keyword called finally?	Exception handling	
14	What are the types of variables java handles?	Data type & variable	65	Java does not provide destructors?	Class & Object	
15	How is object destruction done in java?	Class & Object	66	Do the vector class is contained in java.util package?	Package	
16	What is a string in java?	String	67	Does private modifier can be invoked only by code in a subclass?	Inheritance	
17	What are the different access specifiers available in java?	Package	68	Does all files are included in the java.io package?	Package	
18	What is the default access specifier in java?	Package	69	Java supports multiple inheritance?	Interface	
19	What is a package in java?	Package	70	Do java.applet is used for creating and implementing applets?	Applet	
20	Name some java api packages	Package	71	How applets are programs that executes within a java enabled web browser?	Applet	
21	Explain the features of java language.	Basics	72	Is java a high-level language?	Basics	
22	Compare and contrast java with c.	Basics	73	What are byte codes and java virtual machine?	Basics	
23	Compare and contrast java with c++.	Basics	74	Explain about java variables.	Data type & variable	
24	Discuss in detail the access specifiers available in java.	Package	75	Differentiate between java applications and java applets.	Applet	
25	Explain the different methods in java.util.arrays class with example.	Array	76	What is a thread in java?	Thread	
26	How multiple inheritance is achieved in java?	Package	77	Explain the meaning of public static and void modifiers for the main() method in a java program.	Class & Object	
27	How does java handle integer overflows and underflows?	Data type & variable	78	Explain about inheritance in java.	Inheritance	
28	How java handle overflows and underflows?	Data type & variable	79	Explain about polymorphism in java.	Inheritance	
29	What are the threads will start when you start the java program?	Thread	80	Explain the structure of a java program.	Basics	
30	What is java math class? List 10 method with syntax.	Package	81	What are the steps for implementing a java program?	Basics	
31	Explain java data types?	Data type & variable	82	Explain java data types.	Data type & variable	
32	What is java array?	Array	83	Write the different operators in java.	Operator	
33	Write short notes on java method with syntax and example.	Class & Object	84	What are the control statements available in java?	Control structure	
34	What is java variable? Explain the different types of variable.	Data type & variable	85	What are the looping statements available in java?	Control structure	
35	Explain garbage collection in java.	Class & Object	86	What are the different string methods available in java?	String	
36	There is no destructor in java, Justify.	Class & Object	87	What are the different string buffer methods available in java?	String	
37	What are java classes?	Class & Object	88	What is the use of this keyword in java?	Class & Object	
38	How we can create java classes.	Class & Object	89	What is the use of super keyword in java?	Inheritance	
39	How we can create java objects?	Class & Object	90	What is the use of finally keywords in java?	Exception Handling	
40	What is java string?	String	91	Explain about different class modifiers.	Package	
41	How we can initialize and create java string explains with 10 methods?	String	92	Explain about different constructor modifiers.	Class & Object	
42	Explain java character class with suitable example and methods	Package	93	Explain the use of method modifiers.	Inheritance	
43	What is inheritance in java? Explain all its type with example.	Inheri-tance	94	Write short notes on different java api packages.	Package	
44	Explain interface in java. How do interfaces support polymorphism?	Interface	95	Write short notes on different exception types available in java.	Exception handling	
45	Explain package in java. List out all packages with short description.	Package	96	Explain about catch, throw and try statement in java.	Exception handling	
46	What is java interface.	Interface	97	Why does java not support destructors and how does the finalize method will help in garbage collections?	Class & Object	
47	Explain exception handling in java.	Exception handling	98	Write short notes on access specifiers and modifiers in java.	Package	
48	What led to the creation of java?	Basics	99	Discuss the working and meaning of the “static” modifier with suitable examples.	Class & Object	
49	What are the steps to be followed for executing a java program?	Basics	100	Explain in detail as how inheritance is supported in java with necessary example.	Inheritance	
50	Explain the data types available in java	Data type & variable	101	Explain in detail as how polymorphism is supported in java with necessary example	Inheritance	
51	What are the different types of operators in java?	Operator	102	What are the java apis used for package?	Package	

Figure 8 Returned similar questions belonging to different topics by Jaccard similarity.

Biased to exact word match

While framing a question, keywords and functional words are integrated and sequenced in an appropriate manner to make meaning out of the question. The use of these words by the learner is the natural outcome of the learner's knowledge and communication skill. And as a reason, lack of a learner's expertise does not assure the correctness of question framing. The similarity assessment technique performs an exact word match. This will return only those questions, the words of which are exactly matched (word-by-word) with the learner's input question. This results in obscuring many other similar questions, which are having different words but similar or near to similar meanings. And thus, many of the questions having similar meanings but having different word construction are ignored, resulting in poor efficiency.

Proposed framework for correct question suggestion to the learner

Considering the above-mentioned three problems, we have adopted the soft cosine technique to find similar sentences. The similarity matching is augmented by question selection and iteration pass. We propose a similarity assessment framework for suggesting the correct question for a given incorrect question on a particular domain. The framework consists of three phases of working, as discussed below. The framework is shown in Fig. 9, while the process flow is shown in Fig. 10.

Figure 9 The proposed framework for correct question suggestion to the learner.

Figure 10 The flow diagram for suggesting correct questions to the learner.

Selecting questions with similar concepts

The selection of questions with similar concepts limits the search boundary, and hence the diverse information issue can be addressed. Learners impose questions using the best of their knowledge. This makes them use concepts that are more aligned with the information they are trying to seek. Though not all the concepts which are articulated in the question are rightly chosen, the probability of having the required concept in the question also persists. And thus, claiming all questions from the corpus having the same concept(s) as present in the source question could increase the likelihood of finding the right intended question. This also reduces the probability of recommending questions that are completely on a different topic(s) or concept(s) not relating to the concept(s) present in the source question. As a reason, the concept-wise selection of questions will reduce the diversification of information recommendation.

Similarity assessment and correct question recommendation

A learner may compose an incorrect question due to the following three reasons:There are insufficient keywords to express the question.

Insufficient number of words used to express the question.

The selection of words and their usage may be incorrect.

In all the cases, we need to find the alternative questions closest to the learner's intended question. For estimating the similarity, we suggested looking for the questions that have the same or similar word features as the learner's question. A hard similarity (word to word) match for word features between the incorrect and alternative question reduces the chances of getting a more accurate alternative. Moreover, conducting a hard similarity search in the word feature space of the correct question, the source question's inappropriate words would be of no use. Rather a soft similarity (synonym or close related words) match would give a high probability of finding the questions that are meaningfully aligned to the learner's intent. To address the similarity match problem and to find the correct question, we applied soft cosine measures. Soft cosine allows finding the questions that are significantly similar in terms of the semantic matching, irrespective of the exact word match.

The similarity measure sim (fi, fj) in soft cosine calculates the similarity for synonym or relatedness between the features fi and fj of the vectors under consideration. Here, the vector is a question, and the words of the question represent its features. A dictionary approach like WordNet::Similarity is being used to calculate the similarity (or relatedness) among the features (Sidorov et al., 2014).

From the n-dimensional vector space model's perspective, the soft cosine measures the semantic comparability between two vectors. It captures the orientation (the angle) between the two vectors. But unlike cosine similarity, the features are projected in an n-dimensional space so that similar features are close by with very less angle difference. This causes the meaningfully similar words (features) of vectors (questions) to have minimal angle differences (Hasan et al., 2019), as shown in Fig. 11. The equation for soft cosine is given in Eq. (9).

Figure 11 Comparison between (A) cosine and (B) soft cosine.

(9) Soft_cosine(p,q)=∑i,jN⁡Sijpiqj∑ijN⁡Sijpipj∑ijN⁡Sijqiqj

Where, Sij is the similarity between the features i and j, and p and q are the input question and the correct question, respectively.

Iteration and question selection

To overcome the issue of information overload, ten questions whose similarities are found more than 50% in relation to the source question text are enlisted to choose by the learner. This allows the learner to focus much on what he is actually seeking rather than getting overwhelmed by the huge information which would have been recommended otherwise. Since the approach is probabilistic, chances are there that no right question which is close to learner intention is found in the list. In such a case, selecting a question from the recommended list nearer to the question which learner intends to seek would allow the system to have better-informed data. The learner selected questions that, in turn, act as a seed for further similarity search. Considering the selected question (seed question) as new input for further similarity search would actually converge the search boundary and increase the homogeneity of information. This will reduce diversification. With every recommendation pass, the degree of concept-wise similarity increases, which, in turn, increases the range of similar questions. This makes the question suggestion to shift closer to the learner's intention. The complete process is presented in Algorithm 1.

Algorithm 1 Finding the correct question as per learner intent.

Input:	Incorrect question	Wq	
Corpus	crp	
Output: The intended question	
Label 1: concepts[] = get_concept(Wq)
Selected_question[] = search_question(crp, concepts)
Similar_correct_question[] = soft_cosine_similarity(Selected_question, Wq)

for q in similar_correct_question then
similarity = score_similarity(q)
if similarity > 0.50 then
print q
end if
end for

print "input the question and abort/search"
input q, status

if status == “Abort” then
print q, “is the intended question”
else if
Wq = q
goto Label 1
end if	

Experiment for correct question suggestion

Experimental procedure

For experimentation and performance analysis, the proposed methodology for similarity assessment and recommendation of correct question is implemented as a web-based client/server model, as shown in Fig. 12.

Figure 12 The web (client/server) model used to implement the proposed framework.

The server contains the web application (WebApp) with the requisite HTML and Python file, Flask (https://flask.palletsprojects.com/en/1.1.x/) framework, and Python (version 3.8). Flask is a web application microframework that allows to delegate web pages over the network and handle learner's input requests. The framework is glued as a layer to Python for executing the processes. The model is implemented in Python and is deployed in WebApp as a Python file. Further, the learner's different interactions with the system are stored as the experimental data in the SQLite database, which comes default with Python.

The web server is connected to the client devices over the internet or LAN to exchange HTTP Requests and HTTP Responses. And, the learner (client) interacts with the model through the webpage, as shown in Fig. 13. The reason behind choosing this web model for the experiment is as follows:

Figure 13 User interface for learner interaction.

Python programs allow for text-based interaction, which disinterests learners, making them less attentive. This causes lacking full involvement of the learner. In contrast, a web-based model gives a graphical interface for interaction and thus better involvement of the learner.

Since the experiment involves many learners, a web-based model allows them to participate in the experimentation from anywhere and anytime. This gave the learner more freedom to choose the place and time of their own to take part in the experimentation. Moreover, this web model allows multiple candidates to simultaneously participate in the experiment from different client devices while the experimentation result is getting stored centrally.

Selection of the questions based on the concept, and followed by similarity assessment, is carried out in the server. Three similarity assessment techniques—soft cosine, Jaccard, and cosine similarity are used to find the intended correct questions from the corpus. These three techniques are followed in parallel for assessing their performance for the given incorrect input questions. For this experiment, we used the complete training corpus (i.e., 2,533 questions).

To select the probable correct question from the recommend similarity list, a threshold of 0.5 is considered as the minimum similarity score for soft cosine, while 0.2 is considered for Jaccard and cosine. It was found that Jaccard and cosine similarity techniques returned either no or very few (one or two) similar questions, which were not suitable for carrying out the experiment. Further, in some cases, while searching for similar questions to the given incorrect question, the same question is iteratively returned for each consecutive pass. As a reason, in the cases of Jaccard and cosine, the threshold for similarity score is reduced to a lower value of 0.2. This gave some outputs needed to carry out the experiment and compared to the result of soft cosine.

Learner verification

The performance of the framework for similarity-based recommendation to find the intended question was verified by manual assessment. The assessment was carried by a group of learners. A total of 34 students of the CSE department at Bengal Institute of Technology, studying Java in their 6th semester of the B.Tech degree program, were selected. The students chosen were low scorers in the subject. The rationale behind choosing these students was that we wanted to select learners who are aware of the Java language and its terminology but are neither expert nor good in the subject. This made them suitable candidates as they were susceptible to compose incorrect questions.

Each student was instructed to inputs approximately three incorrect questions, totaling 100. Corresponding to each question, three recommendations are made using the soft cosine, Jaccard, and cosine similarity techniques, as shown in Fig. 13. If the student found the correct intended question, the iteration was stopped for the respective similarity technique. If the intended question was not found in the recommended list, the student chose a question from the list as a seed question that was close to the intended question, and another iteration or pass was followed. If the intended question was not found within three passes, the recommendation process for the respective individual similarity technique was stopped. The purpose of using three similarity techniques is to make a comparison and find the best performance among the three.

Result and analysis

Accuracy

The learner input and feedback on a total of 100 incorrect questions are shown in Table 8. The learner acceptance result of finding the intended correct question against the incorrect input question is summarized and shown in Fig. 14. The summarization is made on the basis of whether the learner finds the intended question or not for each of the three similarities-based recommendations.

Figure 14 Comparing the correct question recommendation based on three similarity metrics: (A) soft cosine, (B) cosine and (C) Jaccard.

Table 8 Similarity recommendation against learner questions.

User Input	Error type	Intended question	Soft Cosine	Cosine	Jaccard	
No. of Pass
(Score)	No. of Pass
(Score)	No. of Pass
(Score)	
what difference interface	IS	What is the difference between abstract class and interface?	1(0.63)	1(0.51)	NF	
define method in subclass with same name	IS	It is not possible to define a method in the subclass that has the same name same arguments and the same return type.	1(0.66)	1(0.5)	NF	
java not have destroy and how garbage collect	EG	Why does Java not support destructors and how does the finalize method will help in garbage collections?	2(0.61)	NF	NF	
how to overload	IS	What is method overloading? Explain with example.	2(0.6)	NF	NF	
why main public	IS	Why is main method assigned as static?	2(0.75)	NF	2(0.6)	
object stored reach	EG	When an object is stored are all of the objects that are reachable from that object stored as well?	1(0.91)	NF	NF	
what mechanism used for a single thread at a time	EG	What is the mechanism defined by java for the resources to be used by only one thread at a time?	1(0.59)	2(0.41)	1(0.36)	
applets talk on web page	IS	How can I arrange for different applets on a web page to communicate with each other?	1(0.57)	NF	1(0.42)	
show try catch throw	IS	Write a Java program which illustrates the try catch throw and throws and finally blocks.	1(0.64)	NF	NF	
why thread synchronization needed	IS	Describe the need of thread synchronization. How is it achieved in Java programming? Explain with a suitable program.	2(0.51)	2(0.33)	2(0.37)	
access modifiers in java	IS	Explain access modifiers and access controls at class and package level in Java.	1(0.58)	NF	1(0.25)	
difference between exceptions	IS	What is difference between user defined exceptions and system exceptions?	NF	1(0.37)	NF	
inbuilt exceptions in class	IS	Explain with example any three inbuilt exceptions and any three inbuilt methods of exception provided by exception class.	1(0.51)	NF	NF	
class extends another class how to handle exception	IS	If my class already extends from some other class then what should I do, if I want an instance of my class to be thrown as an exception object?	1(0.72)	NF	NF	
if we do not initialize variables	IS	What happens if you do not initialize an instance variable of any of the primitive types in Java?	NF	NF	NF	
Inheritance hierarchy in AWT.	EG	Draw the inheritance hierarchy for the frame and component classes in AWT.	1(0.53)	1(0.37)	NF	
which specifier to use while all not interface implement	EG	If you do not implement all the methods of an interface while implementing what specifier should you use for the class?	1(0.75)	NF	1(0.5)	
first value of array elements	IS	What will be the default values of all the elements of an array that are defined as an instance variable?	1(0.56)	NF	1(0.33)	
difference between two types programming language	IS	What is the difference between an object-oriented programming language and object-based programming language?	1(0.68)	1(0.41)	NF	
we change throws when override	EG	Can we modify the throws clause of the superclass method while overriding it in the subclass?	2(0.55)	NF	NF	
name of object with own lifecycle	IS	What is it called where object has its own lifecycle and child object cannot belong to another parent object?	1(0.52)	NF	NF	
boolean value operators	IS	Which of the operators can operate on a Boolean variable?	3(0.57)	NF	1(0.4)	
From main call and check string palindrome or not	IS	Write a method that checks if a string is a palindrome. Call your method from the main method.	1(0.57)	NF	1(0.5)	
methods String available under Name class some. Buffer	ES	What are the different buffer string methods in Java?	1(0.62)	1(0.33)	NF	
high power file copy	IS	Which streams are advised to use to have maximum performance in file copying?	NF	NF	NF	
compare different controls for visibility	IS	Explain the different visibility controls and also compare with each of them.	1(0.54)	1(0.51)	1(0.66)	
use reflection to build array	IS	How to create arrays dynamically using reflection package.	1(0.59)	NF	NF	
voice message with playMessage method	EG	Develop a message abstract class which contains playMessage abstract method. Write a different sub-classes like TextMessage VoiceMessage and FaxMessage classes for to implementing the playMessage method.	NF	NF	NF	
all methods of object class	IS	Explain the different methods supported in Object class with example.	2(0.53)	2(0.33)	2(0.42)	
special style of text example	EG	How do achieve special fonts for your text? Give example.	1(0.64)	1(0.54)	1(0.42)	
keep integer overflow	IS	How does Java handle integer overflows and underflows?	1(0.65)	1(0.47)	NF	
thread start initial	IS	When a thread is created and started what is its initial state?	1(0.56)	1(0.47)	NF	
shift operation in short circuit	EG	Explain short circuited operators and shift operators	1(0.59)	1(0.31)	NF	
what are different interface implement	IS	Describe different forms of interface implementation with their syntax declaration.	NF	NF	NF	
all ways to call method	IS	What are the different ways of calling a static method from a program?	1(0.52)	NF	NF	
java program to create person from class	IS	Consider a class person with attributes firstname and lastname. Write a Java program to create and clone instances of the Person class.	1(0.72)	NF	1(0.4)	
vector difference show	EG	How vector is different from array? Illustrate with programming Example.	2(0.68)	1(0.33)	NF	
break statement how different	EG	Write the difference between break and continue statements in Java.	1(0.56)	1(0.4)	2(0.67)	
more than one inheritance support	EG	What is inheritance? Is multiple inheritance supported by Java?	1(0.52)	1(0.31)	2(0.33)	
applet application program	IS	What is an applet? How do applets differ from an application program?	1(0.82)	NF	1(0.5)	
import class in program	EG	How can class be imported from a package to a program?	1(0.76)	NF	1(0.4)	
can interface be used in class	IS	Is it possible to use few methods of an interface in a class? If so, how?	1(0.66)	NF	NF	
monitor procedure for many	IS	What is the procedure to own the monitor by many threads?	1(0.64)	1(0.44)	1(0.33)	
package import auto	IS	Does java.lang package is automatically imported into all programs.	2(0.72)	1(0.47)	NF	
what is architecture independence	IS	Explain architecture neutral & platform independent.	2(0.51)	NF	2(0.33)	
important classpath variable	EG	Write an importance of classpath variable.	1(0.73)	NF	1(0.4)	
need to import lang package	EG	Do I need to import Java lang package any time? Why?	1(0.81)	NF	1(0.66)	
what is serial	EG	Explain serialization?	1(0.68)	NF	NF	
what locale class	IS	What is the significance of Locale class?	1(0.86)	NF	NF	
what are the alternatives to inheritance	EG	Mention some alternatives to inheritance.	2(0.55)	NF	NF	
is the method what finalize? of use	ES	Explain the use of finalize method	1(0.86)	1(0.57)	1(0.6)	
each of control for what the is use structure	ES	What is the use of each control structure?	1(0.86)	1(0.5)	1(0.6)	
any give 4 ++ C and Java differences. between	ES	Give any 4 differences between Java and C++.	1(0.6)	NF	1(0.25)	
Of what handle can the are Java programs types	ES	What are the different types of program Java can handle?	NF	NF	1(0.42)	
What platform is independency	ES	What is platform independency?	1(0.81)	1(0.57)	1(0.5)	
in of symbols comment types different are java What the	ES	What are the different types of comment symbols in Java	1(0.57)	NF	1(0.5)	
How is a constant in Java defined	ES	How is a constant defined in Java?	1(0.81)	1(0.47)	1(0.4)	
use keyword is the what of final	ES	What is the use of final keyword?	1(0.86)	NF	1(0.6)	
the of is control structure use what each for	ES	What is the use of each control structure?	1(0.86)	1(0.5)	1(0.6)	
constants constants static and compare final	ES	Compare static constants and final constants	1(0.53)	NF	1(0.5)	
need methods for is the what static	ES	What is the need for static method?	1(0.72)	1(0.37)	1(0.6)	
platform supports how java independency	ES	How Java supports platform independency?	1(0.72)	1(0.51)	1(0.5)	
to important java why is internet	ES	Why Java is important to the internet?	1(0.54)	NF	1(0.16)	
is internet to java important why	ES	Why Java is important to the internet?	1(0.57)	NF	1(0.16)	
Application and Applet Compare	ES	Compare applet and application	NF	NF	NF	
difference copy with clone	EG	Differentiate cloning and copying.	2(0.59)	1(0.35)	NF	
pros and cons of static nested class	EG	Write the advantages and disadvantages of static nested class.	1(0.64)	2 (0.32)	1(0.37)	
what fields method	EG	Explain about Final class Fields Methods.	NF	NF	NF	
package access specifier	EG	What do you understand by package access specifier?	1(0.63)	1(0.51)	1(0.33)	
priority in garbage collector	IS	Garbage collector thread belongs to which priority?	1(0.73)	1(0.67)	1(0.6)	
circle filled when right click	IS	Develop Java program that changes the color of a filled circle when you make a right click.	1(0.64)	1(0.56)	1(0.2)	
explain assertion use	IS	What is an assertion? What is its use in programming?	NF	NF	NF	
array fill	IS	Give the syntax for array fill operation.	1(0.63)	NF	1(0.4)	
method to demon thread	IS	Which method is used to create the demon thread?	1(0.51)	NF	NF	
what class on read side byte stream	EG	Name the filter stream classes on reading side of byte stream?	1(0.71)	NF	NF	
what is the use of input stream	EG	What is the functionality of sequence input stream?	1(0.63)	NF	NF	
see if file is hidden or not	EG	How to check if a file is hidden?	1(0.81)	1(0.43)	1(0.5)	
when was the file last modified	EG	How to get file last modified time?	1(0.7)	1(0.5)	1(0.42)	
use of encapsulation	EG	What is the primary benefit of encapsulation?	NF	NF	NF	
most used algorithm in collection	IS	What are common algorithms implemented in Collections Framework?	1(0.57)	1(0.36)	NF	
how is iterator designed	EG	What is the design pattern that iterator uses?	NF	NF	NF	
component size preferred	IS	What is the preferred size of a component?	1(0.86)	1(0.86)	1(0.75)	
read each line of file	EG	How to read file content line by line in Java?	1(0.81)	1(0.63)	NF	
delete a file that is temporary	EG	How to delete temporary file in Java?	1(0.86)	1(0.61)	NF	
calculate factorial of a number	IS	Java program to find factorial of a number using loops	1(0.51)	1(0.33)	1(0.25)	
thread priorities	IS	Discuss about thread groups and thread priorities.	1(0.75)	NF	NF	
java programming	IS	Explain Java programming environment.	1(0.5)	NF	NF	
what are lexical issues	EG	Discuss the lexical issues of Java.	NF	NF	NF	
data type used for arithmeic operators	EG	Which data type can be operands of arithmetic operators?	NF	1(0.36)	NF	
compound assign	IS	What are compound assignment operators?	1(0.56)	1(0.4)	1(0.25)	
use bitwise in boolean	EG	Can bitwise operators be used in Boolean operations?	1(0.66)	1(0.44)	NF	
order for call awt	EG	What is the sequence for calling the methods by AWT for applets?	NF	NF	NF	
cast object explain	EG	What do you understand by casting an object? Explain with the example	1(0.72)	NF	NF	
find size of object	IS	Does Java provide any construct to find out the size of an object?	1(0.7)	NF	NF	
required for try to follow catch	EG	Is it necessary that each try block be followed by a catch block?	1(0.52)	1(0.42)	1(0.33)	
what is precedence rule	IS	Explain precedence rules and associativity concept	1(0.58)	NF	NF	
type upgrade in method overloading	EG	What type promotion has to do with method overloading?	NF	NF	NF	
explain two types of polymorphism	IS	What is run time polymorphism and compile time polymorphism?	1(0.57)	NF	NF	
explain blockingqueue	EG	What do you understand by BlockingQueue?	1(0.64)	NF	NF	
java and internet	IS	Why java is important to the Internet	1(0.81)	1(0.66)	1(0.66)	
Note:

IS, insufficient information; EG, grammatical error; ES, sequential error; NF, not found.

Based on learner input and the system feedback, the framework is evaluated for the accuracy metric. The accuracy is an intuitive performance measure, a ratio of correct observation made to the total observation made. The accuracy is defined in percentage by Eq. (10).

(10) Accuracy=A×100B

Where,A is the number of observations made where the learner finds the correct intended question.

B is the total number of questions taken for observation.

The overall accuracy result of the framework corresponding to the soft cosine, Jaccard, and cosine similarity techniques is shown in Fig. 15.

Figure 15 Accuracy comparison for similar question recommendation of three similarity measures.

The accuracy results for learners accepting the recommended question show that soft cosine similarity outperforms the cosine and Jaccard similarities. In the given experimental data set, the soft cosine based recommendation returns the correct result in two or more passes for 12 input questions. While, for the other 73 input questions, it returns the result in one pass. Therefore, it can be concluded that though the soft cosine similarity-based recommendation returns the intended question in one pass for the maximum number of questions, recommending results in two or more passes is unavoidable. It is observed that input questions lacking sufficient information cause the recommendation system to iterate multiple passes of learner’s interaction to reach the intended question. The hefty size of the corpus might be another reason for the increased number of passes.

The results also show that for 15 input questions, the soft cosine similarity-based recommendation fails to find the correct question matching to learner’s intent. It is observed that in very few cases where the words in the input question are highly scrambled or out of sequence, it may cause the soft cosine to fail to find the correct questions. In this case, the Jaccard similarity outperforms the soft cosine. The other reason which contributes to soft cosine failing is the string length of the input question. If the string length is reduced to one or two words after stopword removal in question preprocessing, the soft cosine based recommendation is unable to find the exact intended question from the huge number of questions within a limited number (three passes) of learner’s interaction. Perhaps a greater number of interactions were needed. Besides these two structural issues on input questions, the soft cosine has some inherent limitation which causes the recommendation set to fail in retrieving the appropriate questions near to learner intention. Even though it is claimed that soft cosine works well on word similarity, actually, it does not do well for multiple synonyms while matching for similarity. The other inherent issue is that the soft cosine fails to infer the common-sense meaning from a sequence of words or phrases to find semantical similarity.

Diversity and evenness

Soft cosine technique with every iteration converges the search for questions on a particular topic. This causes the recommended questions to be very much focused on the intent of the input question. To assess the effectiveness of soft cosine in each pass, the iteration result of the recommended question list, obtained by the three similarity assessment techniques, is analyzed for diversity and evenness. The diversity specifies how the questions in the recommended list are diverse in terms of topic. Where the evenness specifies how evenly the topic information (concepts) are spread (distribution) in the recommended list. The diversity and the evenness of information in the recommended list of questions in each pass are calculated by Shannon's diversity index (H) and Shannon's equitability (EH), respectively, as given by Eqs. (11). and (12).

(11) H=−∑i=0n⁡Piln⁡Pi

Where, n is the number of topic category and Pi is the proportion of the number of ith topic relative to the total count of individual topics for all questions in the recommended list.

(12) EH=Hln⁡S

Where, S is the total count of individual topics for all questions in the recommended list. The evenness value assumes between 0 and 1, where 1 denoting completely even. In an ideal situation, H ≈ 0 specifies that topic in recommendation question list is not diverse and all recommended question focuses on one topic. Similarly, EH ≈ 0 specifies zero dispersion of topics in the recommended question list.

The changes in diversity and equitability indices along to each pass for a given incorrect question “java not have destroy and how garbage collect” are discussed below.Each keyword in the source question denotes a concept which in turn relates to a topic. The keywords in the question are used to select and group questions from the corpus belonging to the same topic domains. The incorrect question is matched with the grouped question using the soft cosine measure. The set of suggested questions returned by the soft cosine similarity measure in the first pass is shown in Table 9. Each keyword in the recommended similar question list reflects a concept which accounts for a count of the respective topics. Based on which the H and EH are calculated for the list as given in Table 10.

The learner chooses the question “explain garbage collection in java programming” from the recommended list of questions which is closest to her intent as the seed question for further searching.

In the second pass, again, based on the keywords from the source question, the questions on the same topic are selected and grouped from the corpus. The set of suggested questions returned by the soft cosine similarity for the selected question against the selected source question is shown in Table 11.

Table 9 Suggested similar questions from first iteration (pass 1).

Pass	Suggested similar question	
Pass 1	Explain garbage collection

How we can create java classes

How we can create java objects

Explain garbage collection in java programming

What is garbage collection

How to create a file in Java

How to read a file in Java

	

Table 10 Diversity and evenness measures from pass 1.

	Basic	Data type & variable	operator	Control structure	Array	String	Class & object	Inheritance	Interface	Package	Exception handling	Thread	Applet	I/O	Total topic count	
x	5	0	0	0	0	0	5	0	0	0	0	0	0	2	12	
p(x)	0.416667	0	0	0	0	0	0.416667	0	0	0	0	0	0	0.166667		
ln(p(x))	−0.87547	0	0	0	0	0	−0.87547	0	0	0	0	0	0	−1.79176		
p(x).ln((px))	−0.36478	0	0	0	0	0	−0.36478	0	0	0	0	0	0	−0.29863		
Diversity	1.028184 (Calculated using Eq. 11)	
Evenness	0.935893 (Calculated using Eq. 12)	

Table 11 Suggested similar questions from first iteration (pass 2).

Pass	Suggested similar question	
Pass 2	What is garbage collection?

Explain java data types?

Explain garbage collection.

Explain garbage collection in java programming.

What is the purpose of garbage collection in Java? When is it used?

Explain finalize and garbage collection in Java

How are objects released in garbage collection?

	

Based on the individual topic count and the total topic count, the H and EH are calculated for the list, as given in Table 12. It is evident that the diversity index H = 1.02 in pass 1 is reduced to H = 0.85 in pass 2. This implies that the diversity of topic information found in the recommended list decreases along with the passes. This signifies the search information space converges, which give learner to be focused and better options to select the question from the list. Further, the evenness EH = 0.985 in pass 1 is reduced to EH = 0.781 in pass 2. This implies that the unevenness of topic distribution among the questions increases. This signifies that the distribution of the intended topic among the question increases which give a high probability of finding the right question.

Table 12 Diversity and evenness measures from pass 2.

	Basic	Data type & variable	Operator	Control structure	Array	String	Class & object	Inheritance	Interface	Package	Exception handling	Thread	Applet	I/O	Total	
x	4	1	0	0	0	0	8	0	0	0	0	0	0	0	13	
p(x)	0.307692	0.076923077	0	0	0	0	0.615385	0	0	0	0	0	0	0		
ln(p(x))	−1.17865	−2.564949357	0	0	0	0	−0.48551	0	0	0	0	0	0	0		
p(x).ln((px))	−0.36266	−0.197303797	0	0	0	0	−0.29877	0	0	0	0	0	0	0		
Diversity	0.858741 (Calculated using Eq. 11)	
Evenness	0.78166 (Calculated using Eq. 12)	

The keyword-based selection and grouping of questions from corpus eliminates the otherwise irrelevant questions and thereby restricts it to a reduced topic search space. Further, soft cosine measure based similarity concretely shrinking the search to more meaningful questions close to the learner’s intent and thereby decreasing the diversity.

From the results, a sample of nine questions that passed two iterations, applying the soft cosine similarity, was considered. Table 13 shows the diversity and evenness calculated on the topic information for the recommended question list obtained after each pass corresponding to the three similarity assessment techniques for a given question. Here, diversity and evenness equating to 0 indicate that the suggested question list belongs to the same topic. Some question searches using the similarity-based technique led the learner to find the intended question in the first pass. This made the second pass for the question search a not applicable (NA) case. From the table, it is quite clear that with every pass, the diversity in the recommended list of the question, obtained by soft cosine in comparison to other, decreases. This made us conclude that with the progression of search iteration, the search space becomes narrower; in other words, the search converges. This ensures the search result to be focused on the intended topic, which helps the learner in reaching the intended question quickly.

Table 13 Diversity index and equitability on recommended questions.

	Soft cosine	Cosine	Jaccard	
Pass 1	Pass 2	Pass 1	Pass 2	Pass 1	Pass 2	
Diversity	Evenness	Diversity	Evenness	Diversity	Evenness	Diversity	Evenness	Diversity	Evenness	Diversity	Evenness	
Java not have destroy and how garbage collect	1.02	0.93	0.85	0.78	0	0	1.72	0.96	0	0	0.41	0.37	
how to overload	0.67	0.97	0.50	0.72	0	0	1.03	0.94	0	0	0.79	0.72	
why thread synchronization needed	0.45	0.41	0	0	0.32	0.46	0.60	0.54	0	0	1.69	0.94	
we change throws when override	0.63	0.91	0	0	0.79	0.72	0	0	0.94	0.85	0.50	0.72	
all methods of object class	1.19	0.86	1.08	0.78	1.16	0.84	1.27	0.92	0.75	0.69	0.85	0.78	
vector difference show	0.56	0.51	0.63	0.91	1.58	0.88	NA	NA	0.50	0.72	1.88	0.96	
package import auto	0.41	0.59	0	0	0	0	NA	NA	0	0	0.90	0.81	
what is architecture independence	1.27	0.71	0	0	1.60	0.89	1.60	0.89	0	0	0	0	
Why main public	0	0	0	0	0.50	0.72	1.24	0.89	0	0	0.45	0.65	

Conclusions and further scope

A lot of emphases are given to developing and structuring the contents so that it can be attractive and motivating to learners. Due to the high-cost factor and difficulty in managing peer-to-peer support, learner-expert based interaction is being less encouraged in online systems. Questions are one of the key forms of natural language interaction with computers which gives the learner an upper hand in interacting with computers more broadly. Composing correct questions is essential from this perspective. A rightly composed question allows a clear understanding of what the learner wants to know. An incorrectly composed question raises ambiguity and diversions, which results in incorrect information. This often misleads the learner. For determining the intent and objective and hence the semantics of the question, it is important to know whether the question is composed correctly to its semantics. Determining whether the input question is incorrectly or rightly composed would increase the accuracy of information retrieval. This put the absolute requirement for verifying whether the question framing is and by semantics is correct or not before it can be used for information retrieval.

This paper proposes an approach for assessing the validity of framing the question and its semantics. A tri-gram based language model is used for assessing the question's correctness in terms of syntax and semantics. The model outperforms the other n-gram approaches and establishes the fact that tri-gram optimally performs well in assessing the questions. The tri-gram language model exhibits an accuracy of 92%, which is way higher than the accuracy shown by 2-gram, 4-gram, and 5-gram over the same test data assessment.

The work also proposes an interactive framework for correct question recommendation. The framework uses a soft cosine based similarity technique for recommending the correct question to the learner. The proposed framework is assessed by learner questions and compared with other similarity assessment techniques, viz. cosine and Jaccard. The soft cosine similarity technique recommends the correct question way better than the other two, achieving an accuracy of 85%. In the case of multi-pass interaction, as the number of passes increased, the information diversity is reduced, and the search is converged to the intended question quickly.

In conclusion, incorporating the presented work in an interactive OLS will not only improve the performance of the system significantly but will also enhance the learner satisfaction and learning focus, leading to a boosted quality of learning. The proposed approach can be used in precise personalized learning recommendations and mitigating the associated cold start problem.

However, this work has a couple of limitations which opens up further research scopes. Since we used a tri-gram based approach, it cannot assess the correctness of a question that has less than three words. Also, it fails to assess the informal questions that typically comprise compound and multiple sentences. Techniques like graphs (semantic network), machine learning (LSTM), etc., can be explored to solve these issues.

Supplemental Information

Supplemental Information 1 Python code for finding similarity between user input question and questions from the corpus.

Click here for additional data file.

Supplemental Information 2 Python code for assessing the correctness of the user questions.

Click here for additional data file.

Supplemental Information 3 Database.

Question corpus

Click here for additional data file.

Additional Information and Declarations

Competing Interests

Author Contributions

Data Availability

The authors declare that they have no competing interests.

Saurabh Pal conceived and designed the experiments, performed the experiments, analyzed the data, performed the computation work, prepared figures and/or tables, authored or reviewed drafts of the paper, and approved the final draft.

Pijush Kanti Dutta Pramanik conceived and designed the experiments, analyzed the data, prepared figures and/or tables, authored or reviewed drafts of the paper, and approved the final draft.

Aranyak Maity conceived and designed the experiments, performed the experiments, analyzed the data, performed the computation work, authored or reviewed drafts of the paper, and approved the final draft.

Prasenjit Choudhury analyzed the data, authored or reviewed drafts of the paper, and approved the final draft.

The following information was supplied regarding data availability:

The corpus used in the experiment and the raw codes are available in the Supplemental Files.

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
