# Peer review of "Learner question’s correctness assessment and a guided correction method: enhancing the user experience in an interactive online learning system"

_PeerJ Computer Science, doi:10.7717/peerj-cs.532_

## Round 0.1 · original submission · Major Revisions

However, the paper can be enhanced by sticking with the following changes.

- A more comprehensive and clearer conclusion is expected.

- The paper can be further improved and proper analysis can be shown to prove the strength of the approach.

- More description of the technical details will help to improve the quality.

- Improving the presentation to emphasize the author's goal will help to improve the quality of the final paper in the final camera-ready version.

- Including some discussions with existing method which could prove the proposed method's effectiveness will improve the quality of the paper.

- The approach can be discussed with some other existing techniques.

- Sentences / English polishing will help to improve the quality of the final paper in the final camera-ready version.

- The references in this manuscript are somewhat out-of-date. Include more recent research in this field.

Reviewer 1 ·

Basic reporting

The first thing I noticed in the paper is its main objective, which is not specific and so
general to study. As well, the researcher does not determine the scope of the study. For
example: In this paper, propose a novel method to assess the correctness of the user query, in terms of syntax and semantics. The question is: “Where is this paper conducting?”.

Secondly, the researcher did not mention any of the references at the end of the paragraph in the first chapter (introduction), and it is so important to mention the name followed with the year of the reference, which these information are taken from it. For example: Online learning systems (OLSs) have brought great advantages to all kind of formal and informal
learning modes. Over the years, OLSs have evolved from simple static information
delivery systems to interactive, intelligent, and context-aware learning systems, virtually
incorporating real-life teaching and learning experience. In today's OLSs, much of the
emphasis is given on designing and delivering learner-centric learning, in terms of the
learning style, learning approaches, and progress of a particular learner.

Experimental design

1-A shortage of information and data regarding the participated sample conducted in this
study including its sample number or any specific information related to them. I did not
find that in the methodology section. And in the abstract section, the researcher only briefly mentions it. For example: A trigram language model is built and trained for assessing the correctness of learners' queries on Java.
2-The researcher does not explain the web model (client or server) well, as this model is used in the experimental procedure in this paper. For example, the proposed framework for the correct question is implemented as a web model (client/server) for experimentation and performance analysis. And this paper also does not explain the reasons behind choosing this model for this paper.
3-Moreover, There are concepts in this paper that do not have a specific definition or explanation that makes this paper hard to understand and appears too complex to read. Such as the Softcosine technique. This technique does not have enough information about how it works, and how to benefit from it to achieve the paper objective.

Validity of the findings

1-A lot of significant information mentioned in this paper but without a clear and specific
sequence, which makes the paper hard to understand and the information are not found
effectively. And that means there is no benefits from most of the results of this paper.
2-The research questions, objectives do not clearly formulate in this paper.
3-No discussion section for the results found and comparing to what was discovered in the
previous studies regarding the paper issue.

Additional comments

The main aim of the paper is strong and recent and has a strong significance in this field. It will add value to the educational system. The experimental analysis used is fully explained and effectively evaluated the results of the study. In addition, having a lot of tables and charts that explained the results in a clearer and detailed way.
But, you have to fix the paper according to the following comments:
1-The whole paper should be rearranged with specific main titles and sub-titles. And each title must completely explain all the significant information that may benefit in this section.
2- The research question, objective, and sub-objectives and (hypothesis of the research paper if needed) should formulate.
3- More in-depth information about the introduction and background of the research issue in the first chapter should be mentioned, to give the reader more information and data about the research issue.
4- The sample method and model used in this research paper should discuss and explain in what way can be beneficial in achieving the research paper objectives.
5- It should mention all the models used in this paper in the methodology section and the framework of this research paper should be explained in more detail by determining the dependent and in-depended variables for the research paper.
6- The researcher must discuss the results with the previous outcomes found in past related studies.
7- It is so important to determine the scope of the study to make the research more beneficial.
8- All references must be cited in the paper, if there is a piece of information it must be cited for easy checkout.
9- It must formulate some recommendations before the conclusion section, to contribute in improving the educational system using the results found in this paper.

Long sentences:
• Page 6 line 52: “Over the years, OLSs have evolved from simple static information delivery systems to interactive, intelligent, and context-aware learning systems, virtually incorporating real-life teaching and learning experience.”

• Page 6 line 61: “To achieve advanced learning skills like analyzing, evaluating, creating, and applying, a higher level of interactions like discussion, hands-on experiments, exchanging views with experts, etc. are required.”

• Page 16 line 399: “Further analysis of these false-negative questions reveals that most of them after preprocessing and stopword removal the question length is reduced to less than three.”

• Page 16 line 417: “Similarly decreasing N leads to word sequence pattern search at lower order, and this restricts the probability of correctness of the word sequences at higher orders. This typically lowers accuracy.”

• Page 18 line 459: “Failing to determine which parts of the question and thus their semantics are aligned to learner's intent lead to ambiguity in identifying the parts of a sentence to be taken correctly for similarity matching.”

Better words:
• Page 6 line 57: “one key aspect of an OLS is interactivity”
o One key aspect of an OLS is interaction.

• Page 6 line 58: “But, despite the advantages, due to high-cost factor and complexity, contents developed for OLSs have limited or no interactivity.”
o Limited or no interaction.

• Page 6 line 66: “The best option is to opt for a question-answer based OLS”
o The best option is to choose a question-answer based OLS.

• Page 7 line 73: Fundamentally, these systems process the input query to parse its structure and semantics to understand the intention of the query.
o Fundamentally, these systems process the input query to determine its structure and semantics to understand the intention of the query.

• Page 7 line 82: “For instance, more often than not, the non-native English-speaking people having poor knowledge of English find it difficult to compose queries in English.”
o Most often.

• Page 9 line172: Error checking in the text is a problem since long back.
o Error testing the text has been a problem for a long time ago.

• Page 14 line 345: “The correctness of a question is estimated based on its syntactical and semantic aspects and accordingly is classified as correct or wrong.”
o As right or wrong.


World abbreviation:
• Page 9 line 158: what’s POS refer to?

Citation:
• It need more citation, there is no citation on most of the paper.

• Page 9 line 196: “Pattern recognition is one of the successful ways of detecting errors in the textual sentence. In [3], a learning model is generated from labelled sequential patterns of both correct and wrong sentences.”
o better citation.

• Page 10 line 205: “To detect the word usage error bidirectional long short-term memory (LSTM) model is proposed in [5].”
o better citation.

• Page 17 line 448: “Cosine and Jaccard similarity techniques are the two text-based similarity approach which has been widely incorporated for finding similar text.”
o Need citation.

·

Basic reporting

Improve English:

1) Who is "her" in this paragraph?
"The proposed model has exhibited 92% accuracy while assessed on the test data. Furthermore, in case the query is not correct, we also propose an approach to guide the user leading to a correct question complying her intent."

2) Not recommended to use "more often than not" in a scientific paper.

3) In the context of this paper: "interactivity" is not clear and fuzzy. Provide a coherent definition. "Like every learning process, one key aspect of an OLS is interactivity,"

4) "Insufficient domain knowledge also leads to frame a wrong question" INSTEAD OF "Insufficient domain knowledge also leads to wrong framing of a question."

5) The title of this section should be rephrased "1.2 Plausible Way Outs and their Limitations". It is not suitable for a scientific paper.

Experimental design

Add references:

1) This section "1.2 Plausible Way Outs and their Limitations" is describing some limitations in NLP and Pattern matching. However, no references that support these limitations.

2) "1.3 Proposed Solution" needs to be designed based on facts supported with references.


3) It would be interesting to add references in 2020

4) Is "Table 1" a research result of another research paper? If yes , then add the reference. If no, then explain how this table was found. Example: did the author conduct a content analysis? what is the methodology so that the results in this table can be supported.

Validity of the findings

no comments

Additional comments

no comments

Reviewer 3 ·

Basic reporting

-in the the abstract there is a confusion of terms like method, approach, or framework.
-In the abstract there is an ambiguity in the word "Java".
-abstract does not include the source of dataset used in the paper.
-the problem statement and the objective is not defined clearly.
- the missing of the phase model training of the proposed model in the proposed method.
- the measure of the correctness of the query is not enough and not clear.

Experimental design

-in line 299 the sentence "we corrected a total number of 2533 questions of core-Java from
300 different websites and books" ; it is not clear how the questions are corrected.
- in line 351, the phrase "backof model"; there is no such model in the litrature!.
-in the section 4.3 line 525 to 533 the justification of the using of soft-cosine similarity is unclear.
-the authors must refer to the following reference "Soft similarity and soft cosine measure: Similarity of features in vector space model"
-section 4.4 : there is no justification of using thresholds 0.5, 0.2, 0.2 for soft-cosine , Jaccard and Cosine respectively.
- table 8, line 658 ; need more justifications and discussions on how the passes changes the values of the diversity and evenness measures. and the meaning of these values.

Validity of the findings

no comment

---

## Round 0.2 · Minor Revisions

All comments have been considered into the revised paper.

Reviewer 1 ·

Basic reporting

No comment

Experimental design

No comment

Validity of the findings

No comment

Additional comments

Paper format:
- Text alignment seems to be off.
- The references citation doesn't appear in order, it seems like the ones missing were just injected in the middle without considering the sequence although it is considered in the rest of the paper.
- In the appendix, table 4 caption says "Table 1. Performance measures of the proposed approach".
- Sub-sections title font like the one in line 122 (1.3.1) seems to be too small and unrecognizable which makes it hard to follow through and identify the beginning of the section.

Content:
- I still believe the paper lacks consideration for future work to improve the proposed framework or other further work recommended for e-learning experiment enhancement.

Language:
- Minor grammatical mistakes like missing articles such as in line 333 (an error instead of error), 343 (should be a/the user instead of user), line 906 ( the same instead of same), 905 (the corpus instead of corpus), and many others.
- Grammatical mistakes like subject/verb disagreement like in line363 ( should be the system gives instead of giving) Also (this puts instead of put.
- Line 966 says "Questions or questions", I believe that is meant to be "questions and answers". Also, line 968 says question or question, not sure if that's intended but it doesn't make sense to me.
- Wrong usage of [pre[osions such as in line 883 (should be along with instead of along to).

---

## Round 0.3 · accepted · Accept

The paper can be accepted since the authors improved the papers in terms of structure, presentation and results.